# 2 kW Dual-Output Isolated DC/DC Converter Based on Current Doubler and Step-Down Chopper

**Yoshinori Matsushita** [1,2,*] **, Toshihiko Noguchi** [1]**, Noritaka Taguchi** [2] **and Makoto Ishii** [3]

[1]  Graduate School of Science and Technology, Shizuoka University, Hamamatsu 432-8561, Japan; noguchi.toshihiko@shizuoka.ac.jp

[2]  Yazaki Research and Technology Center, YAZAKI Corporation, Susono, Shizuoka 410-1194, Japan; noritaka.taguchi@jp.yazaki.com

[3]  In-Vehicle Systems R & D Center, YAZAKI Corporation, Susono, Shizuoka 410-1194, Japan; makoto.ishii@jp.yazaki.com

*  Correspondence: matsushita.yoshinori.15@shizuoka.ac.jp

**Abstract:** In the context of the auxiliary power for motor-driven vehicles having two systems, we propose a new topology for a dual-output isolated DC/DC converter, which offers advantages in terms of efficiency and size. The proposed circuit consists of an H-bridge inverter, a transformer, and an integrated circuit of a current doubler and step-down chopper. Considering the high power and high frequency, our objective was to evaluate and identify the issues of an actual device with a power output of 2 kW and switching frequency of 400 kHz. The circuit feasibility was examined through measurements of the prototype, and both the voltage target response and load disturbance response characteristics were confirmed to operate as designed. The maximum and minimum efficiencies of this circuit were 81.3 and 61.5%, respectively, demonstrating that the load loss of the step-down chopper had a significant impact on the efficiency. The loss analysis revealed that the loss at the integrated circuit on the secondary side accounted for more than 50% of the total loss. Moreover, issues such as the behavior at power-on, efficiency, and size were identified and evaluated, thereby achieving the objectives of the study.

**Keywords:** dual output; DC/DC converter; auxiliary power source; current doubler; step-down chopper

## 1. Introduction

Approximately one quarter of the global $CO_2$ emissions, which are the cause of global warming, originate from the transportation sector [1]. To reduce such emissions, the automotive industry has introduced motor-driven vehicles such as electric vehicles (EVs), plug-in hybrid vehicles (PHVs), and fuel cell vehicles (FCVs) into the market, the sales of which continue to increase [2]. In September 2020, the state of California announced a plan to ban the sale of new gasoline-driven vehicles in 2035. Accordingly, the sale of motor-driven vehicles is expected to accelerate in the future. Isolated DC/DC converters are used as auxiliary power for these motor-driven vehicles, with the power supply for the motor drive as the input [3,4]. The capacity and power consumption of auxiliary power is expected to increase in the future in response to the electrification of traditionally mechanical functions, such as steering and reclining mechanisms, as well as the increase in comfort-enhancing electrical components such as seat heaters [5,6]. The increased capacity leads to an increase in the conduction loss, and changing the conventional 12 V auxiliary power to 48 V has been proposed to reduce this loss [7,8]. However, as loads that require a 12 V power supply still exist, the mixing of the two load systems, namely 48 and 12 V, continues to be necessary. At present, several methods are available to realize the two systems' auxiliary power; for example, installing two isolated DC/DC converters

with the power supply input for the motor drive or inserting a non-isolated bidirectional DC/DC converter for 48 and 12 V in a 48 V system [9]. However, these methods exhibit problems such as an increased size owing to the installation of two units and reduced efficiency as a result of the two power conversions. To overcome these issues, we propose a novel dual-output isolated DC/DC converter, which includes an integrated circuit of a current doubler and step-down chopper on the secondary side and outputs both 48 V and 12 V [10]. Since the secondary side of the proposed circuit is an integrated circuit and there is no need to convert voltage twice to generate 12 V, the proposed circuit is expected to have a smaller size and higher efficiency although the control method becomes more complex. As our previous studies were based on simulations or a downsized prototype, no actual device testing has yet been performed with power for automotive applications.

Meanwhile, a demand exists for the downsizing of power converters, and one of the methods to achieve this goal is increasing the frequency. In this manner, the passive components in circuits, such as the transformer, inductance, and capacitance, can be downsized. Therefore, substantial research has been conducted on high-frequency drive power converters using switching devices made of GaN, which enable faster switching than Si devices [11–14]. However, as the frequency increases, the loss at the downsized passive components and switching loss of the switching device also increase, leading to a larger cooling mechanism. Therefore, a specific frequency exists to realize the minimum device size. This frequency is dependent on the input/output power conditions, but it is expected to increase with the improvement in the structures and materials of the passive components [15–17].

Within this context, we conducted a study on a novel dual-output isolated DC/DC converter with an integrated circuit of a current doubler and step-down chopper on the secondary side, which was previously only evaluated for downsizing. We constructed an actual device with a maximum output power of 2 kW for automotive applications, examined its validity, and identified its issues. In consideration of the trend of higher frequency as explained above, a switching frequency is set to 400 kHz, which is about two times higher than switching frequencies of existing studies on a three-port-isolated DC/DC converter [18–20].

## 2. Circuit Configuration and Operation Principle

### 2.1. Circuit Configuration

Figure 1 presents the proposed circuit, consisting of a dual-output isolated DC/DC converter, with an input voltage $V_{in}$ and two output voltages $V_{out1}$ and $V_{out2}$. The input/output isolation is realized by a high-frequency transformer. An H-bridge inverter consisting of S1–S4 is connected on the primary side of the transformer. A current doubler consisting of L11, L12, D1, D2, and C1 (Figure 2a) as well as an interleaved step-down chopper consisting of S5, S6, D1, D2, L21, L22, and C2 (Figure 2b) are connected on the secondary side of the transformer to form an integrated circuit. D1 and D2 are included in both circuits, thereby reducing the number of components. The H-bridge inverter and high-frequency transformer have conventional simple configurations, and the two outputs are achieved solely with the secondary side circuit.

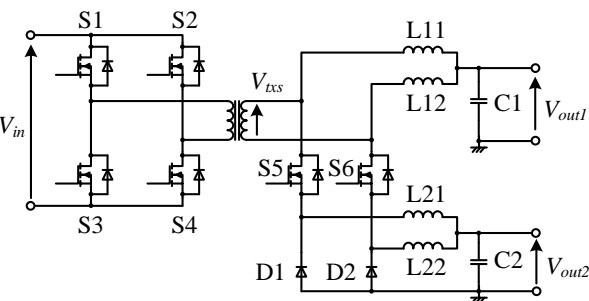

**Figure 1.** Proposed dual-output DC/-DC converter.

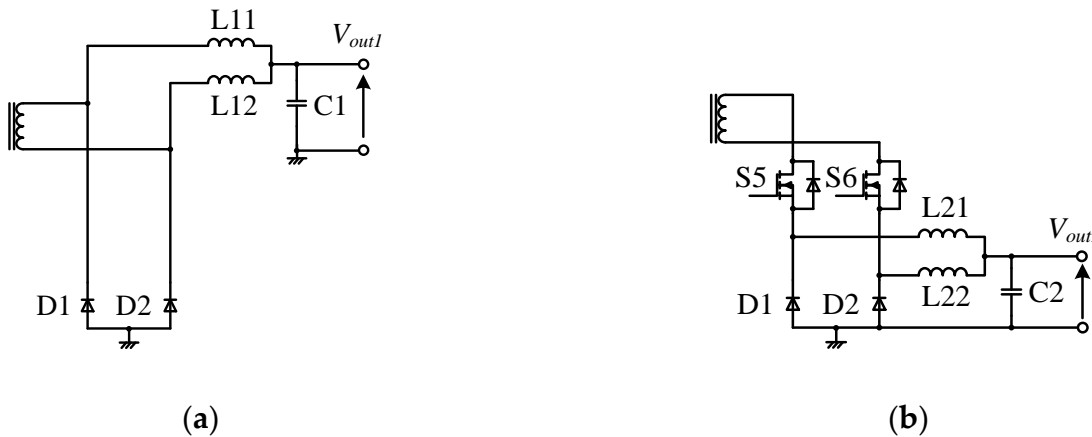

**Figure 2.** Decomposed configuration of secondary circuit: (**a**) current doubler; (**b**) step-down chopper.

The current paths L11 and L12 are for the current doubler output, and the current paths L21 and L22 are for the step-down chopper output. As a result, the conduction loss is reduced by half compared to a single-load current path. Because the phases of these currents are interleaved with a shift of 180°, the ripple frequencies at C1 and C2 become twice the inverter frequency, which reduces the capacity and thereby the volume of C1 and C2 by half. To maintain the interleaving symmetry, the inductance values of L11 and L12 and of L21 and L22 should be set to the same values.

### 2.2. Operation Principle

By switching S1 to S4, the H-bridge inverter generates three levels of voltage, namely $+V_{in}$, 0, and $-V_{in}$, and these voltages are applied on the primary side of the transformer. This circuit includes six operation modes based on the combination of the transformer secondary side voltage level $V_{txs}$, which is transformed according to the turns ratio *N1:N2*, and the on–off states of S5 and S6.

Table 1 presents the states of $V_{txs}$, S5, and S6, as well as the corresponding operation modes. In Table 1, "1", "0", or "−1" for $V_{txs}$ represent a voltage of a positive value, 0, or a negative among the three levels, respectively. Moreover, "-" for S5 and S6 indicates that it is irrelevant whether it is on or off. The reason for this irrelevance is that in these cases, a current flows from the source to the drain in S5 or S6; the current passes through the body diode even in the off state.

**Table 1.** Operation modes of proposed circuit.

| Operation Mode | 1 | 2 | 3 | 4 | 5 | 6 |
|---|---|---|---|---|---|---|
| $V_{txs}$ | 1 | 1 | 0 | −1 | −1 | 0 |
| S5 | off | on | - | - | - | - |
| S6 | - | - | - | off | on | - |

To explain each operation mode, Figure 3 depicts the ideal waveforms in the main part of the secondary side circuit in steady-state operation, whereas Figure 4 presents the equivalent circuit and current flow of each operation mode in the secondary side circuit. From the top, Figure 3 shows the transformer secondary side voltage $V_{txs}$, the switching states of S5 and S6, the inductor currents on the current doubler side $I_{L11}$ and $I_{L12}$, and the inductor currents on the step-down chopper side $I_{L21}$ and $I_{L22}$.

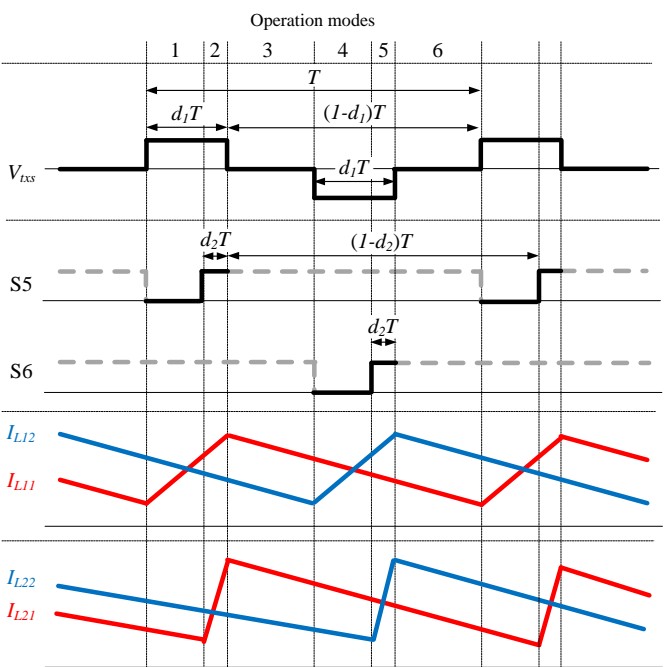

**Figure 3.** Ideal waveforms of each operation mode.

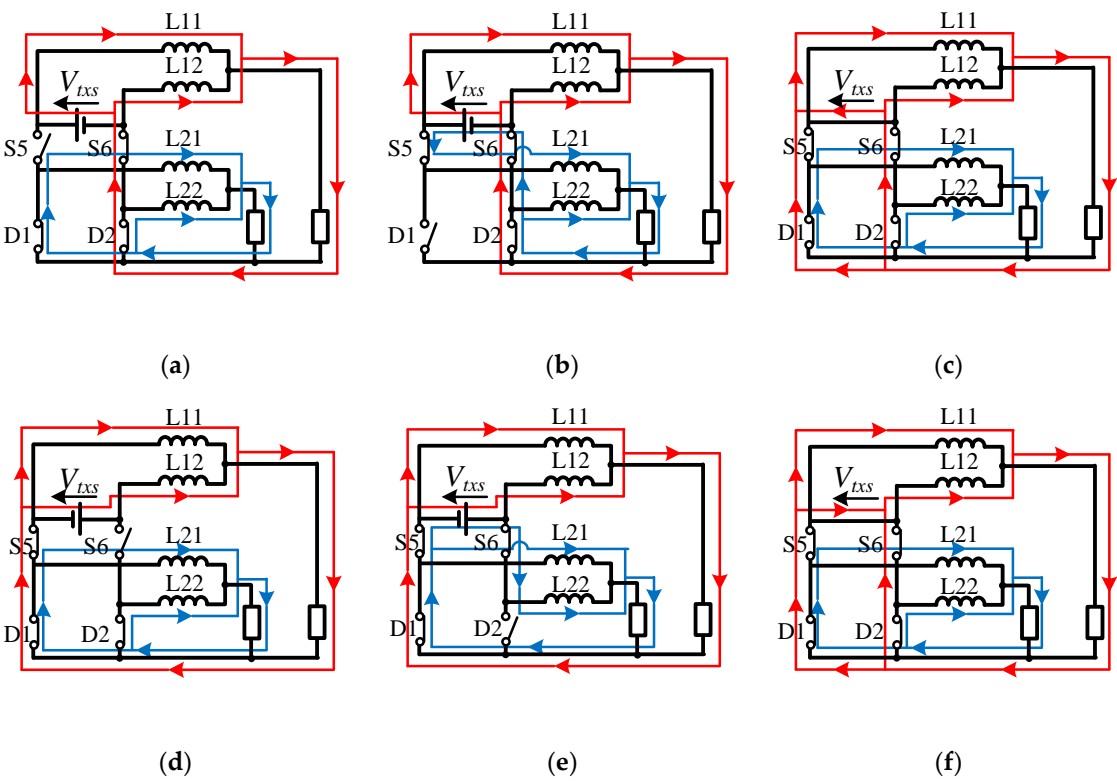

**Figure 4.** Equivalent circuit and current flow of each operation mode in secondary circuit: (**a**) Mode 1; (**b**) Mode 2; (**c**) Mode 3; (**d**) Mode 4; (**e**) Mode 5; (**f**) Mode 6.

The operation of the secondary side circuit in each mode is described below.

- Mode 1: $V_{txs}$ is positive and S5 is off (Figure 4a).

L11 is charged by $V_{txs}$ and L12, L21, and L22 discharge.

- Mode 2: $V_{txs}$ is positive and S5 is on (Figure 4b).

As S5 is turned on, both L11 and L21 are charged by $V_{txs}$, whereas L12 and L22 still discharge.

- Mode 3: $V_{txs}$ is 0 (Figure 4c).

When $V_{txs}$ is 0, there is no power, and therefore, L11, L12, L21, and L22 all discharge.

- Mode 4: $V_{txs}$ is negative and S6 is off (Figure 4d).

L12 is charged by $V_{txs}$ and L11, L21, and L22 discharge.

- Mode 5: $V_{txs}$ is negative and S6 is on (Figure 4e).

As S6 is turned on, L12 as well as L22 are charged. L11 and L21 still discharge.

- Mode 6: $V_{txs}$ is 0 (Figure 4f).

As in Mode 3, as $V_{txs}$ is 0, L11, L12, L21, and L22 discharge.

Table 2 displays the output voltages of the proposed circuit in the steady-state operation $V_{out1}$ and $V_{out2}$; the output currents $I_{out1}$ and $I_{out2}$; the output average inductor currents $\overline{I_{L11}}$, $\overline{I_{L12}}$, $\overline{I_{L21}}$, and $\overline{I_{L22}}$; the inductor current ripple factors $\gamma_{L11}$, $\gamma_{L12}$, $\gamma_{L21}$, and $\gamma_{L22}$; and the capacitor ripple currents $\Delta i_{c1}$ and $\Delta i_{c2}$. In this case, $d_1$ and $d_2$ in the equations are duty ratios for the period in which $V_{txs}$ is positive or negative and for the period of Mode 2 or Mode 5, respectively, as illustrated in Figure 3. $L_1$ and $L_2$ are the inductance values of L11 or L12 and L21 or L22, respectively. $R_1$ and $R_2$ are the load resistance values of the current doubler and step-down chopper outputs, respectively. Moreover, $T$ is the time of one cycle.

**Table 2.** Theoretical formulas of proposed circuit.

| Symbol | Value | Symbol | Value |
|---|---|---|---|
| $V_{out1}$ | $\dfrac{d_1 N_2 V_{in}}{N_1}$ | $V_{out2}$ | $\dfrac{d_2 N_2 V_{in}}{N_1}$ |
| $I_{out1}$ | $\dfrac{d_1 N_2 V_{in}}{N_1 R_1}$ | $I_{out2}$ | $\dfrac{d_2 N_2 V_{in}}{N_1 R_2}$ |
| $\overline{I_{L11}}$ $\overline{I_{L12}}$ | $\dfrac{d_1 N_2 V_{in}}{2 N_1 R_1}$ | $\overline{I_{L21}}$ $\overline{I_{L22}}$ | $\dfrac{d_2 N_2 V_{in}}{2 N_1 R_2}$ |
| $\gamma_{L11}$ $\gamma_{L12}$ | $\dfrac{2 R_1 (1 - d_1) T}{L_1}$ | $\gamma_{L21}$ $\gamma_{L22}$ | $\dfrac{2 R_2 (1 - d_2) T}{L_2}$ |
| $\Delta i_{c1}$ | $\dfrac{R_1 (1 - d_1) T I_{out1}}{L_1}$ | $\Delta i_{c2}$ | $\dfrac{R_2 (1 - d_2) T I_{out2}}{L_2}$ |

As indicated in Table 2, when $V_{in}$ is constant, $V_{out1}$ and $V_{out2}$ are only dependent on the duty ratio corresponding to each of these. That is, the switching of S1 to S4 controls $V_{out1}$, and that of S5 and S6 controls $V_{out2}$. This means that $V_{out1}$ and $V_{out2}$ can be controlled independently; however, this is only possible when the circuit operates with the six operation modes described above. This means that the following condition must be satisfied:

$$d_1 > d_2. \tag{1}$$

If Equation (1) is not true ($d_1 \leqq d_2$), S5 and S6 are always on, while the transformer is transmitting power from the primary side to the secondary side (when $V_{txs}$ is positive or negative). Consequently, there will be no means of controlling $V_{out2}$ and the dual-output converter cannot perform its function.

## 3. Actual Device Testing

### 3.1. Main Circuit Configuration

Table 3 summarizes the specifications of the experimental circuit that was constructed to operate the proposed circuit at 400 kHz and 2 kW, the configuration and appearance of which are presented

in Figures 5 and 6, respectively. The input voltage was 300 V, the input capacitor was a 330 μF aluminum electrolytic capacitor, the output voltage on the current doubler side $V_{out1}$ was 48 V, and the output voltage on the step-down chopper side $V_{out2}$ was 12 V. The switching frequency was 400 kHz, as described previously. For the FPGA to generate control signals, XC7K70T-1FBG484C (Xilinx) with a clock cycle of 5 ns (200 MHz frequency) was used to obtain a sufficient time resolution for switching at 400 kHz (2.5 μs per cycle) to be obtained. A SiC MOSFET SCT3030AL (ROHM) with a short switching transition time was used for S1 to S6 considering the switching loss at 400 kHz. A SiC Schottky barrier diode FFSH4065A (ON Semiconductor) with an extremely short reverse recovery time was used for D1 and D2. Owing to the large current value in the secondary side circuit, S5, S6, D1, and D2 were connected in parallel and a snubber circuit was provided in light of the surge voltage in the off state. The transformer turns ratio was set to 4:2. The inductance values on the current doubler sides L11 and L12 were set to 15 μH, whereas those on the step-down chopper side L21 and L22 were set to 3 μH, considering the relationship among the ripple current amplitude, wire diameter, and core size. The output capacitor on the current doubler side C1 was set to 44 μF by connecting two 22 μF ceramic capacitors in parallel, whereas that on the step-down chopper side C2 was set to 188 μF by connecting four 47 μF ceramic capacitors in parallel. Because the experimental circuit was constructed with the aim of operating at 400 kHz and 2 kW, the selected switching devices as well as the designed transformers and inductors had plenty of room in terms of the breakdown voltage, current capacity, and saturation magnetic flux density.

**Table 3.** Parameters of experimental circuit.

| Parameter | Value |
| --- | --- |
| $V_{in}$ | 300 V |
| $C_{in}$ | 330 μF |
| $V_{out1}$ (current doubler side) | 48 V |
| $V_{out2}$ (step-down chopper side) | 12 V |
| Switching frequency | 400 kHz ($T$ = 2.5 μs) |
| FPGA | XC7K70T-1FBG484C (Xilinx) |
| Clock frequency of FPGA | 200 MHz |
| S1, S2, S3, S4, S5, S6 | SCT3030AL (ROHM) |
| D1, D2 | FFSH4065A (ON Semiconductor) |
| Transformer turns ratio | $N_1$:$N_2$ = 4:2 |
| L11 and L12 | 15 μH |
| L21 and L22 | 5 μH |
| C1 | 44 μF |
| C2 | 188 μF |

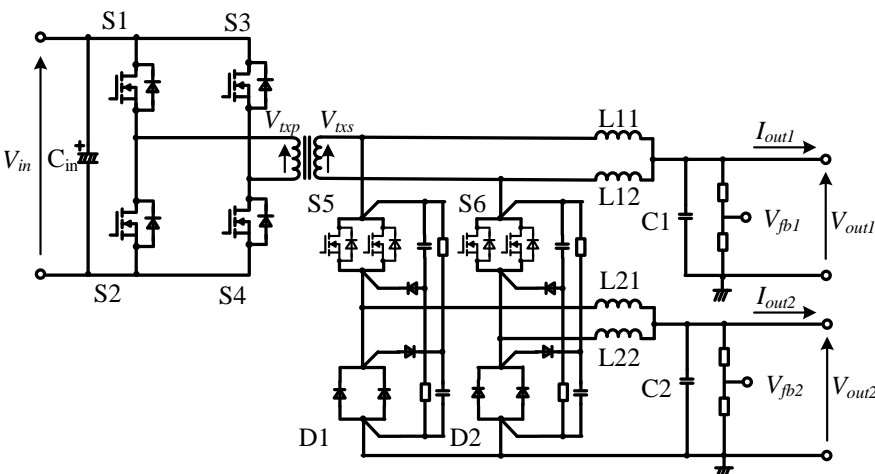

**Figure 5.** Configuration of experimental test circuit.

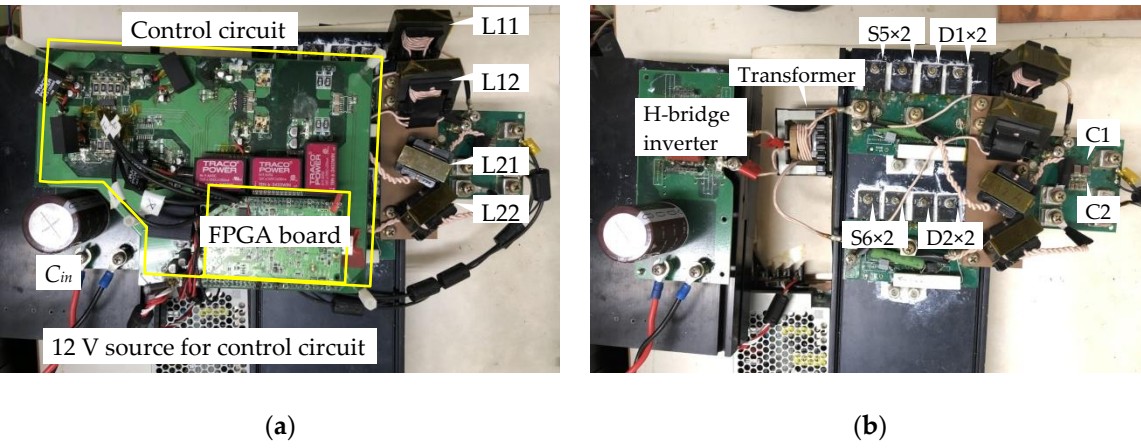

(**a**)　　　　　　　　　　　　　　　　　　　　　　　(**b**)

**Figure 6.** Experimental circuit setup: (**a**) main circuit with control circuit; (**b**) main circuit without control circuit.

### 3.2. Control Circuit Configuration

Figure 7 depicts the control block diagram of the proposed circuit. All switching signals for S1 to S6 were generated by the FPGA (Appendix A), provided with appropriate dead time, and subsequently input to each gate via the isolated gate driver.

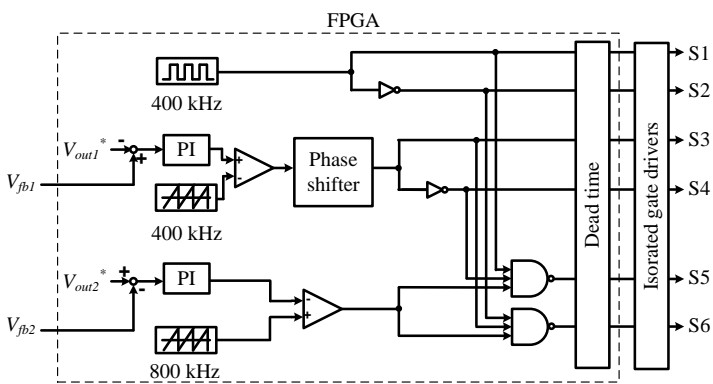

**Figure 7.** Control block diagram of proposed circuit.

A square wave with a duty ratio of 0.5 and a frequency of 400 kHz was input to the gate of S1 and its inverted signal was input to the gate of S2. A phase-shifted signal of the S1 gate signal was input to the gate of S3 and its inverted signal was the gate signal of S4. This phase shift was quantified by the feedback of the output voltage on the current doubler side $V_{out1}$. In particular, the deviation between the divided voltage of $V_{out1}$, denoted by $V_{fb1}$, and the voltage command value $V_{out1}*$ was input to the PI calculation unit. Subsequently, the output of the PI calculation unit was compared with the 400 kHz sawtooth wave synchronized with the S1 gate signal. The time corresponding to the pulse width of the comparison result was the phase-shift quantity.

The gate signals of S5 and S6 were derived from the feedback of the output voltage on the step-down chopper side. Similar to the process for the gate signals of S3 and S4, the first step was to calculate the difference between the divided voltage of the output voltage on the step-down chopper side $V_{out2}$, denoted by $V_{fb2}$, and the voltage command value $V_{out2}*$. Thereafter, the PI-calculated signal of this difference was compared with the 800 kHz sawtooth wave synchronized with the S1 gate signal. In this case, by setting the frequency of the sawtooth wave to twice the switching frequency, the pulse signal after comparison simultaneously generated the interleaved S5 and S6 switching signals. Subsequently, the inverted AND (NAND) of this pulse signal and the H-bridge inverter switching

signal was created to enable the assigned switching of S5 when $V_{txs}$ was positive and that of S6 when $V_{txs}$ was negative, as demonstrated in Table 1 and Figure 3. For the modes in Table 1, in which the switching states of S5 and S6 were irrelevant, S5 and S6 were turned on to prevent loss owing to a current flow to the body diode.

### 3.3. Operation Points

Figure 8 presents the measured operation points and Table 4 displays the measurement instruments. There were five operation points for the output power on the current doubler side $P_{out1}$ between 200 and 1400 W using an inductive resistance of 8 Ω. There were four operation points for the output power on the step-down chopper side $P_{out2}$ between 150 and 500 W using a non-inductive resistance of 1 Ω. These 20 operation points in total were numbered from #01 to #20. An additional operation point #21 was set for an output of 2 kW, with output power on the current doubler side of 1421 W and output power on the step-down chopper side of 588 W. Based on these operation points, the measurements were performed as described in the following, where the numbers shown in Figure 8 are used to indicate the operation points.

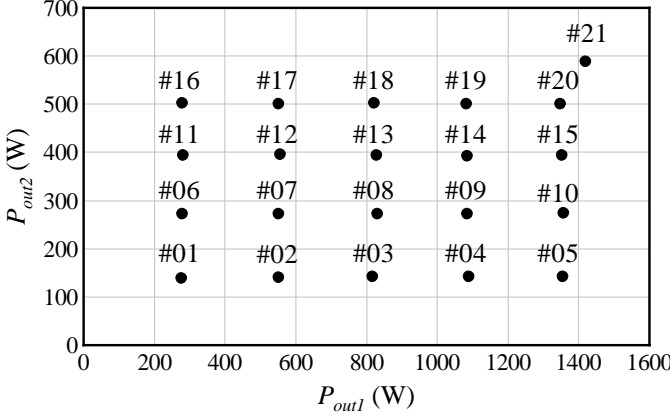

**Figure 8.** Measured operation points.

**Table 4.** Measurement instruments.

| Instrument | Model Number |
|---|---|
| Oscilloscope | HDO6104A-MS (TELEDYNE) |
| Voltage differential probe | 700924 (YOKOGAWA) |
| Current probe (<30 A) | TCP312A (Tektronix) |
| Current probe (>30 A) | TCP303 (Tektronix) |
| Deskew calibration source | DCS025 (TELEDYNE) |

### 3.4. Voltage Target Value Response Characteristics

Figure 9 displays the output voltages $V_{out1}$ and $V_{out2}$, and the output currents $I_{out1}$ and $I_{out2}$ from power-on to the steady state at operation point #21 with a total output power of 2058 W. As indicated in Figure 9, both the voltage and current took less than 1 s from power-on to settling, demonstrating the ability to control two different voltages at an output of 2 kW. The settled voltage values were 48.9 V for $V_{out1}$ and 12.4 V for $V_{out2}$. These were shifted from their command values of 48 and 12 V, which was caused by an error in the voltage dividing resistance ratio. In the steady state, the ripple voltage was ±1.3 V for both $V_{out1}$ and $V_{out2}$, whereas the ripple current was ±0.8 A for $I_{out1}$ and ±1.6 A for $I_{out2}$. The reason for the larger value of the ripple current $I_{out2}$ compared to $I_{out1}$ is that TCP303 (Tektronix), which was used to measure $I_{out2}$, has a lower resolution than TCP312A (Tektronix), which was used to measure $I_{out1}$.

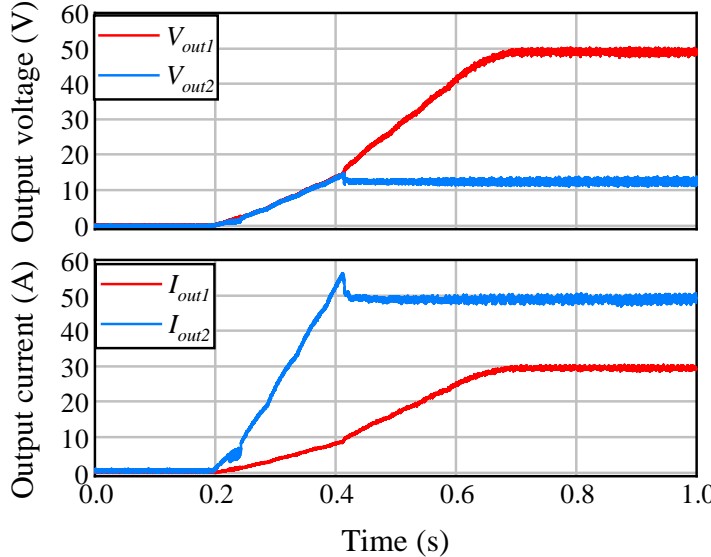

**Figure 9.** Dual-port output waveforms.

In this experiment, the DC power supply was powered on after starting the control, and in this manner, the time to settle the voltage was constrained by the rise time of the DC power supply. That is, the proportional gain and time constant of the PI control unit, which were control parameters, were not involved in the behavior during power-on. These control parameters were optimized for the load disturbance response, as described later. If this circuit is used as on-board auxiliary power, the control will start with the battery voltage applied to the input. Accordingly, the time to reach a steady state will be shortened compared to that in this experiment, and the current and voltage overshoots are expected to be larger. To prevent this, countermeasures such as soft-start are necessary, which we consider as future tasks.

### 3.5. Operating Waveforms

Figure 10 presents the waveforms at four operation points during the steady-state operation: $V_{txp}$, $V_{txs}$, $I_{txp}$, $I_{txs}$, $V_{gs5}$, $V_{gs6}$, $V_{ds5}$, $V_{ds6}$, $V_{diode1}$, $V_{diode2}$, $I_{L11}$, $I_{L12}$, $I_{L21}$, and $I_{L22}$. $V_{txp}$ is the transformer primary side voltage, $I_{txp}$ and $I_{txs}$ are the transformer primary and secondary side currents, $V_{gs5}$ and $V_{gs6}$ are the gate-source voltages of S5 and S6, $V_{ds5}$ and $V_{ds6}$ are the drain-source voltages of S5 and S6, and $V_{diode1}$ and $V_{diode2}$ are the voltages across D1 and D2, respectively. In our setup, the oscilloscope had four channels and could not measure all waveforms simultaneously. Therefore, one channel was used to measure the $I_{txp}$ waveform constantly as the trigger channel to synchronize the results of multiple measurements. The measurements were performed at four operation points: #01, where both $P_{out1}$ and $P_{out2}$ were minimized; #05, where only $P_{out1}$ was maximized based on #01; #16, where only $P_{out2}$ was maximized in the same manner; and #20, where both $P_{out1}$ and $P_{out2}$ were maximized. As can be observed from Figure 10, the operation proceeded at 2.5 μs (400 kHz) per cycle at all four operation points and there were indeed six operation modes, as described in Table 1 and Figure 3. These results confirm that the proposed circuit behaved as designed. Meanwhile, surges and associated ringing appeared in the secondary side circuit voltage at all operation points. The ringing frequencies were approximately 15 and 60 MHz, and these were considered as resonances owing to the transformer leakage inductance and the parasitic capacitance of S5 and S6 and D1 and D2, respectively. Noise was also observed at these frequencies in $V_{gs5}$ and $V_{gs6}$, indicating that high-frequency noise propagated in the entire secondary side circuit.

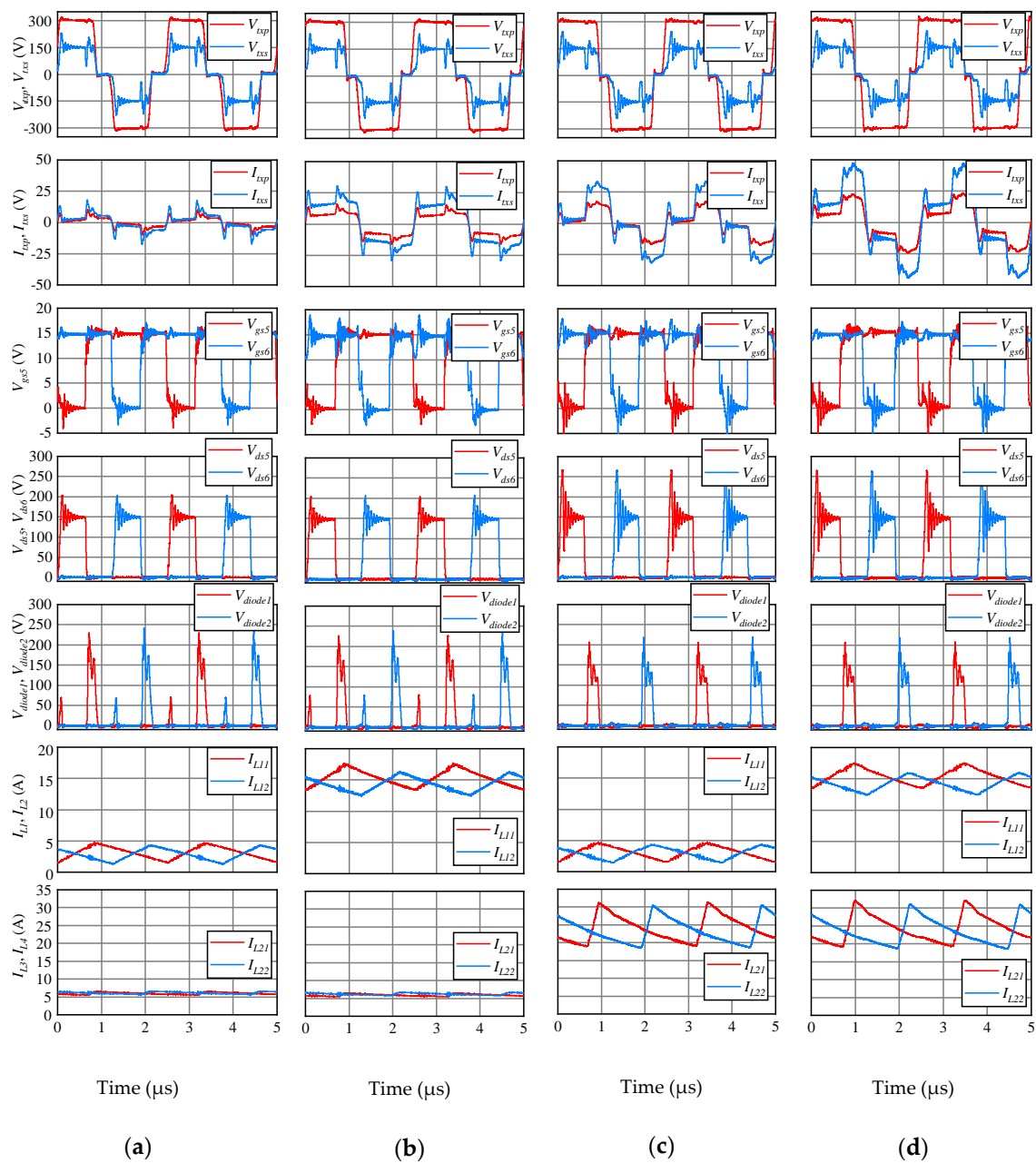

**Figure 10.** Waveforms of proposed circuit: (**a**) #01; (**b**) #05; (**c**) #16; (**d**) #20.

### 3.6. Detailed Operation Modes

To explain the detailed steady-state operation, Figure 11 presents the waveforms of $V_{txp}$, $V_{txs}$, $I_{txp}$, $I_{txs}$, $V_{gs5}$, $V_{gs6}$, $V_{ds5}$, $V_{ds6}$, $V_{diode1}$, and $V_{diode2}$ for a half period at three operation points, namely #01, #05, and #16, in an enlarged view. This half period corresponded to the ideal operation modes 1, 2, and 3, as previously described in Table 1, Figures 3 and 4. Although the behavior varied slightly depending on the operation point, there were nine modes for the measured waveforms in total, which are divided by $t_n$ ($n = 1$ to 11) in Figure 11. In addition to the ideal operation modes 1, 2, and 3, six additional modes for state transitions were present. Table 5 displays the possible detailed operation modes for each operation point, and Figure 12 depicts the current path of the secondary side circuit in each operation mode. The operation in each mode is described below based on Figures 11 and 12.

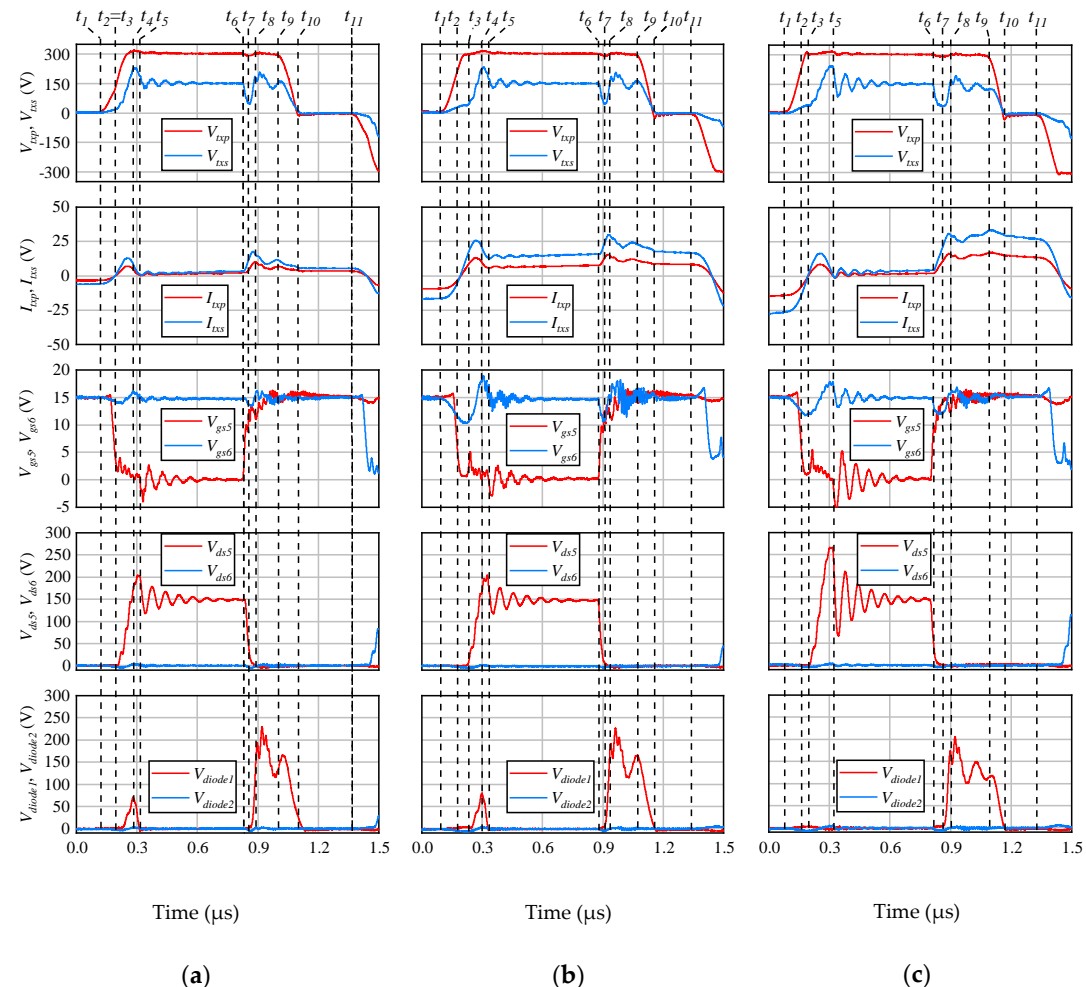

**Figure 11.** Waveforms of proposed circuit for half period: (**a**) #01; (**b**) #05; (**c**) #16.

**Table 5.** Detailed operation modes.

| Operation Point | Operation Mode | | | | | | | | | | | |
|---|---|---|---|---|---|---|---|---|---|---|---|---|
| | **1** | **2-1** | **2-2** | **3-1a** | **3-1b** | **3-2** | **4** | **5** | **6** | **7** | **8** | **9** |
| | $t_1 \sim t_2$ | $t_2 \sim t_3$ | | | $t_3 \sim t_5$ | | $t_5 \sim t_6$ | $t_6 \sim t_7$ | $t_7 \sim t_8$ | $t_8 \sim t_9$ | $t_9 \sim t_{10}$ | $t_{10} \sim t_{11}$ |
| #01 | ○ | | | ○ | ○ | | ○ | ○ | ○ | ○ | ○ | ○ |
| #05 | ○ | ○ | | ○ | ○ | | ○ | ○ | ○ | ○ | ○ | ○ |
| #16 | ○ | | ○ | | | ○ | ○ | ○ | ○ | ○ | ○ | ○ |

1.  ($t_1 < t < t_2$) Period in which $V_{txp}$ began to increase (Figure 12a).

    With the switching of the primary side inverter, $V_{txp}$ started to increase. Furthermore, S5, S6, D1, and D2 were in the on-state. The currents flowed from the source to the drain in S5 and from the drain to the source in S6, and a negative current flowed in the transformer. Because both ends of the transformer were connected to GND, a voltage was generated at $V_{txs}$ corresponding to *L di/dt* of the current path, but not corresponding to $V_{txp}$.

2.  ($t_2 < t < t_3$) Period in which current flowed to S5 body diode (Figure 12b,c).

    $V_{gs5}$ was turned off and the current flowing to S5 was commutated to the body diode. This mode continued until the current in the body diode became zero. Accordingly, a larger S5 current (larger output power) meant that this mode would last longer. In the case of #01, where $V_{gs5}$ was turned off,

no current flowed from the source to the drain in S5; therefore, this mode did not exist for #01. In the cases of #05 and #16, owing to the different operation points, the commutated S5 body diode current flowed in different paths (Figure 12b,c).

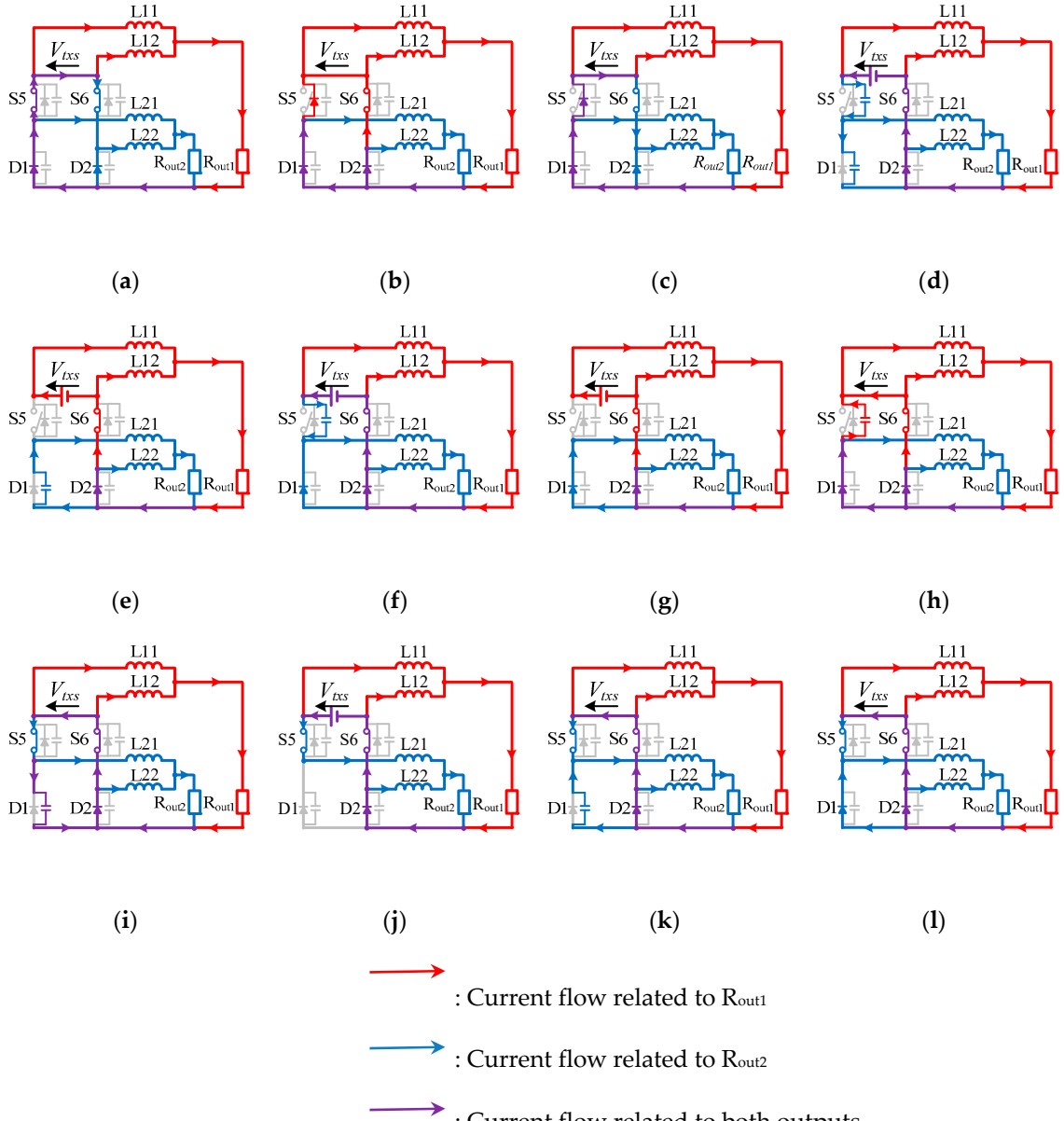

**Figure 12.** Current path of experimental circuit in each operation mode; (**a**) Mode 1; (**b**) Mode 2-1; (**c**) Mode 2-2; (**d**) Mode 3-1a; (**e**) Mode 3-1b; (**f**) Mode 3-2; (**g**) Mode 4; (**h**) Mode 5; (**i**) Mode 6; (**j**) Mode 7; (**k**) Mode 8; (**l**) Mode 9.

Furthermore, in this mode, both ends of the transformer were connected to GND, and thus, no voltage corresponding to $V_{txp}$ was generated at $V_{txs}$ yet. Therefore, when the duty ratio was constant for $V_{txp}$, when this mode was longer (the output power was larger), the duty ratio $d_1$ for $V_{txs}$ was smaller. Because $d_1$ was proportional to the output voltage $V_{out1}$ (Table 2), it attempted to take a constant value when $V_{out1}$ was constant. As a result, the duty ratio for $V_{txp}$ increased as the output power increased.

3.  $(t_3 < t < t_5)$ Charging period for S5 parasitic capacitance $C_{ds5}$ (Figure 12d–f).

The S5 body diode current became zero and $V_{txs}$ started to rise. This $V_{txs}$ acted as a power source to charge $C_{ds5}$ (Figure 12d,f). This charging current was the peak current of $I_{txs}$ and the current change generated a surge voltage at $V_{ds5}$. As described later, this noise propagated to the entire secondary side circuit, including $V_{gs5}$ and $V_{gs6}$. A larger $P_{out2}$ resulted in a larger rate of change in the current. Therefore, the peak surge voltage was greater for #16 than for #01 and #05. Moreover, because the L21 current (D1 current) in the previous mode was small for #01 and #05, the charging current for $C_{ds5}$ flowed to not only L21, but also to D1, causing the charging ($t_3 < t < t_4$) and discharging ($t_4 < t < t_5$) of the D1 parasitic capacitance $C_{D1}$ (Figure 12d,e). This charging and discharging generated a voltage of approximately 70 V at $V_{diode1}$. However, for #16, the D1 current in the previous mode was sufficiently large, and thus, the charging current for $C_{ds5}$ did not charge $C_{D1}$ (Figure 12f).

4.  ($t_5 < t < t_6$) Period corresponding to ideal operation mode 1 (Figure 12g).

    S5 was off and D1 was on, and it operated in mode 1 of the ideal operation principle.

5.  ($t_6 < t < t_7$) Discharging period for $C_{ds5}$ (Figure 12h).

    Discharging of $C_{ds5}$ was caused by turning on $V_{gs5}$, when the current flowing to L21 started to switch from the D1 current to the S5 current. The current continued to flow into S5 and D1 until S5 was on and completely switched; therefore, both ends of $V_{txs}$ were connected to GND again, reducing $V_{txs}$.

6.  ($t_7 < t < t_8$) Charging period for $C_{D1}$ (Figure 12i).

    Once the discharging of $C_{ds5}$ was complete and S5 was completely on, the current started to flow to L21 via S5. At the same time, the charging of $C_{D1}$ started to turn off D1. The peak for $I_{txs}$ was the charging current. The current change at this point generated a surge voltage at $V_{diode1}$. The periods $t_6$ to $t_8$ were the time from the start of discharging $C_{ds5}$ until the end of charging $C_{D1}$. The current involved in the charging and discharging was dependent on $P_{out2}$. Consequently, #16, with a greater $P_{out2}$, had a longer $t_6$ to $t_8$ period than #01 and #05.

7.  ($t_8 < t < t_9$) Period corresponding to ideal operation mode 2 (Figure 12j).

    D1 was completely off and it operated in mode 2 of the ideal operation principle.

8.  ($t_9 < t < t_{10}$) Discharging period for $C_{D1}$ (Figure 12k).

    By switching the primary side inverter, $V_{txp}$ and $V_{txs}$ decreased simultaneously. Accordingly, the current flowing from the transformer to L21 decreased. To compensate for this current loss, D1 was turned on. Discharging of $C_{D1}$ occurred as a preliminary step.

9.  ($t_{10} < t < t_{11}$) Period corresponding to ideal operation mode 3 (Figure 12l).

    D1 was completely on, and it operated in mode 3 of the ideal operation principle.
    The following half cycle behaved symmetrically with the nine modes described thus far. The behavior at each operation point during steady-state operation has been described with the series of operation modes above.

### 3.7. Load Disturbance Response Characteristics

Figures 13 and 14 present the actual measurement results and simulation results of $V_{out1}$, $I_{out1}$, $V_{out2}$, and $I_{out2}$ when the operation point was switched during steady-state operation. Figure 15 shows the waveforms of $V_{txs}$, $V_{out1}$, and $V_{out2}$ at operation point #08. In Figure 13, operation points #02 and #03 were switched to switch $P_{out1}$ only. In Figure 14, #03 and #08 were switched to switch $P_{out2}$ only. PSIM version 12.04 (Powersim) was used for the simulation. The parameters in Table 3 were used in the simulation, and an ideal device was assumed for the switching devices. Furthermore, no parasitic

resistance, inductance, or capacitance existed in the simulation circuit. Figures 13 and 14 verify the following: (1) when switching $P_{out1}$ and switching $P_{out2}$, it recovered to a steady state within 4 ms after the switching of the operation point, and (2) each waveform from the operation point switching to the recovered steady state was in good agreement with the simulated waveform. Based on these results, the load variation control was realized as designed. Compared to the simulation results, the actual measurement results exhibited greater superimposed noise in the voltage and current. These noises were in accordance with the surge timing of $V_{txs}$, as illustrated in Figure 15. As indicated above, a $V_{txs}$ surge was generated at the switching timing of S5, S6, D1, and D2, thereby demonstrating that the switching noises in the secondary side circuit propagated to the output voltage. This indicates the noise propagation via paths that could not be absorbed by the output smoothing capacitors C1 and C2, revealing the layout design issue of the secondary side circuit.

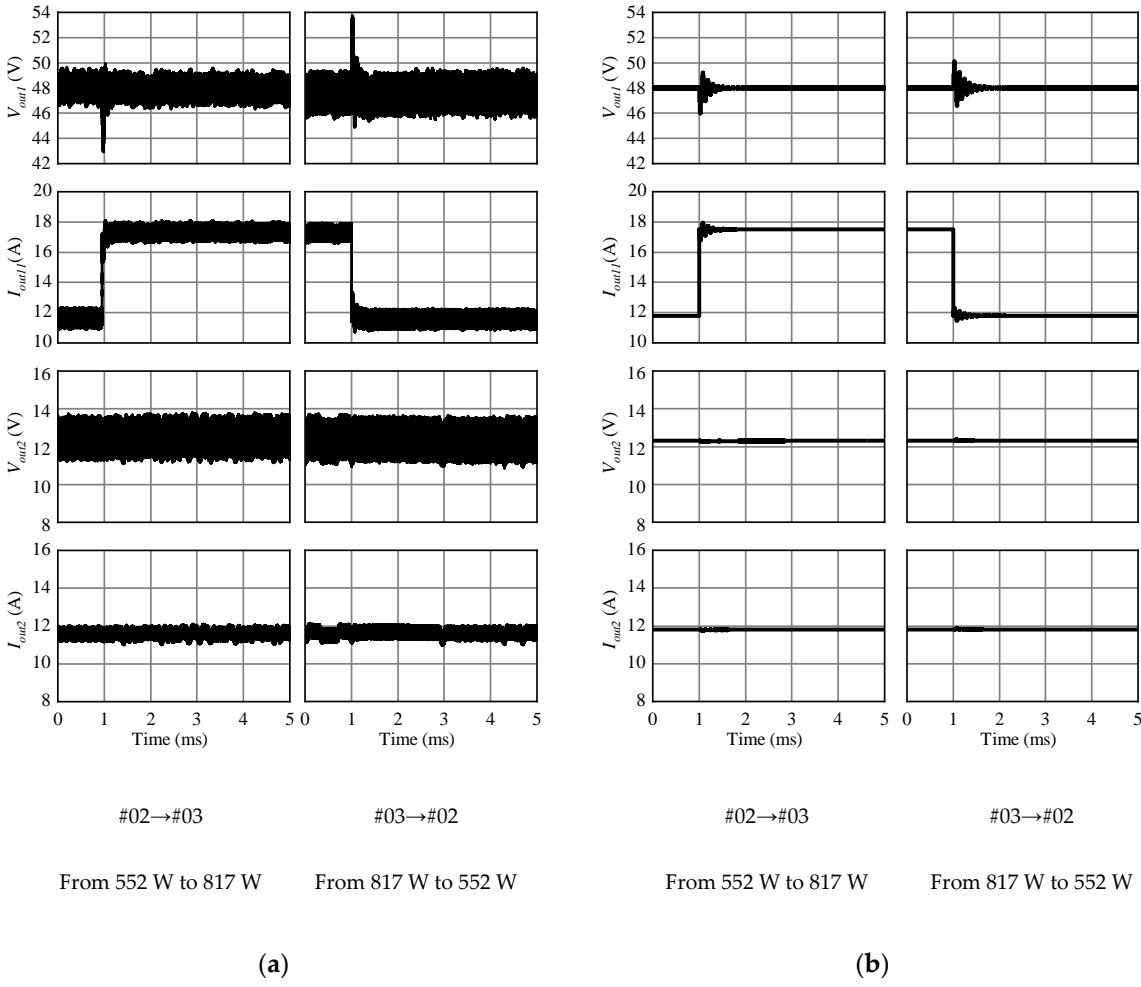

**Figure 13.** Results of $P_{out1}$ dynamic response test when $P_{out2}$ = 141 W: (**a**) measurement results; (**b**) simulation results.

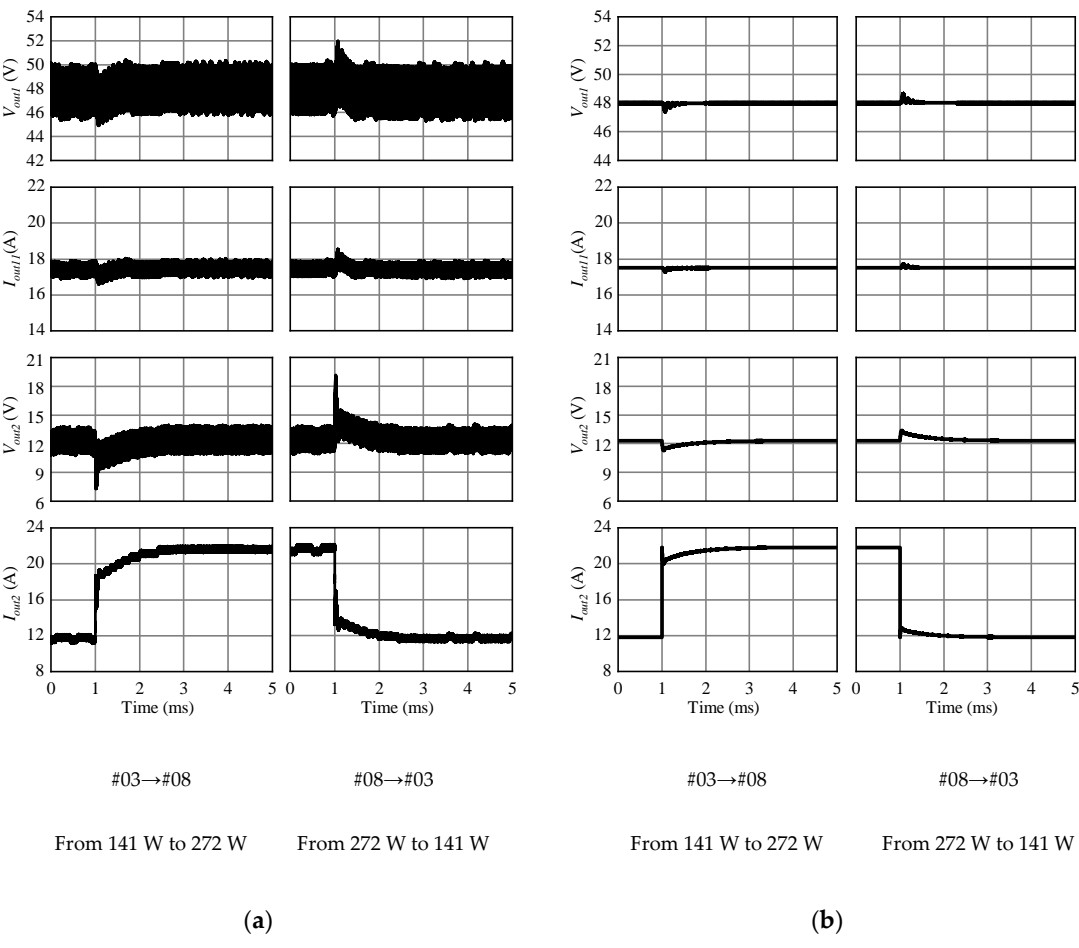

#03→#08  #08→#03  #03→#08  #08→#03

From 141 W to 272 W  From 272 W to 141 W  From 141 W to 272 W  From 272 W to 141 W

(**a**)  (**b**)

**Figure 14.** Results of $P_{out2}$ dynamic response test when $P_{out1}$ = 825 W: (**a**) measurement results; (**b**) simulation results.

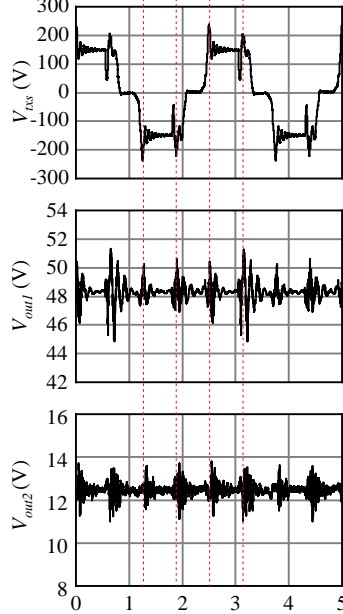

**Figure 15.** Experimental waveforms of $V_{txs}$, $V_{out1}$, and $V_{out2}$ at #08.

The measurement results of the voltage target value response and load disturbance response characteristics verified the behavior of the proposed circuit as designed. Thus, despite certain issues, the 400 kHz, 2 kW dual-output DC/DC converter was demonstrated.

## 4. Efficiency and Loss Evaluation

### 4.1. Efficiency Characteristics

The input power $P_{in}$ and output power $P_{out1}$ and $P_{out2}$ were measured using a power analyzer, namely WT1800 (YOKOGAWA). The efficiency $\eta$ was measured at each operation point of the test circuit based on the following equation:

$$\eta = \frac{P_{out1} + P_{out2}}{P_{in}}. \tag{2}$$

The measurement results are presented in Figure 16. The efficiency increased as $P_{out1}$ increased, whereas it decreased as $P_{out2}$ increased. The maximum efficiency was 81.3% at #05, where $P_{out1}$ was the maximum and $P_{out2}$ was the minimum. The minimum efficiency was 65.1% at #16, where $P_{out2}$ was the maximum and $P_{out1}$ was the minimum. These results demonstrate the significant influence of the load loss of $P_{out2}$ on the reduced efficiency.

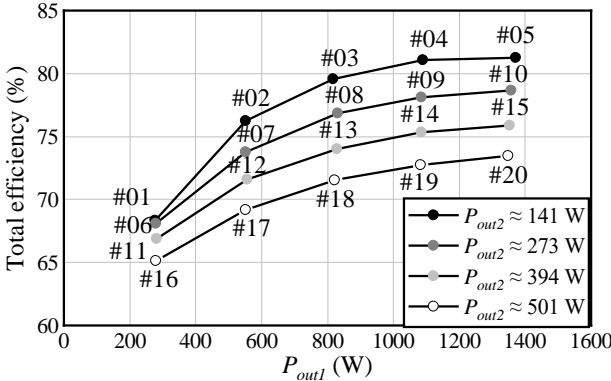

**Figure 16.** Results of efficiency measurements.

### 4.2. Loss Characteristics

The voltage and current of the transformer on the primary and secondary sides were measured using the devices in Table 4 for 100 μs. Moreover, the power $P_{txp}$ and $P_{txs}$ of the transformer on the primary and secondary sides, respectively, were calculated based on the following equation:

$$P = \frac{1}{40T} \sum_{0}^{40T} v(t)i(t)\Delta T, \tag{3}$$

where $P$ is the power, and $v(t)$ and $i(t)$ are the measured voltage and current, respectively. Furthermore, $T$ is the time of one cycle and $\Delta T$ is the time resolution of the oscilloscope with the values of $T = 2.5$ μs and $\Delta T = 0.1$ ns. The calculated $P_{txp}$ and $P_{txs}$ as well as the measured $P_{in}$, $P_{out1}$, and $P_{out2}$ were used to divide the total loss $W_{total}$ into three parts: the inverter loss $W_{inv}$, transformer loss $W_{tx}$, and secondary side loss $W_{sec}$. The derivation formula for each loss is displayed in Table 6. Owing to the convenience of the measurement system, the time of the data acquisition by the power analyzer and the oscilloscope was not simultaneous but differed in each case.

The results of the loss separation are illustrated in Figure 17. The subjects of the loss separation were operation points #01, #05, #16, and #20, which had the maximum or minimum $P_{out1}$ or $P_{out2}$, respectively. The left axis indicates the power loss, and the right axis is the square of the total output

current $I_{out1} + I_{out2}$. The right axis is provided to indicate the conduction loss. As illustrated in Figure 17, the total loss and those at the different locations increased in the order of #01 < #05 < #16 < #20. This trend was consistent with the square of the total output current. Thus, it was demonstrated that the conduction loss was dominant compared to the iron loss or switching loss at the 400 kHz operation. Moreover, the secondary side loss accounted for more than half of the total loss at all operation points. Therefore, to improve the efficiency of the test circuit, it is effective to reduce the conduction loss in the secondary side circuit.

**Table 6.** Loss calculations.

| Part | Symbol | Equation |
|------|--------|----------|
| Total loss | $W_{total}$ | $P_{in} - (P_{out1} + P_{out2})$ |
| Inverter loss | $W_{inv}$ | $P_{in} - P_{txp}$ |
| Transformer loss | $W_{tx}$ | $P_{txp} - P_{txs}$ |
| Secondary side loss | $W_{sec}$ | $W_{total} - (W_{inv} + W_{tx})$ |

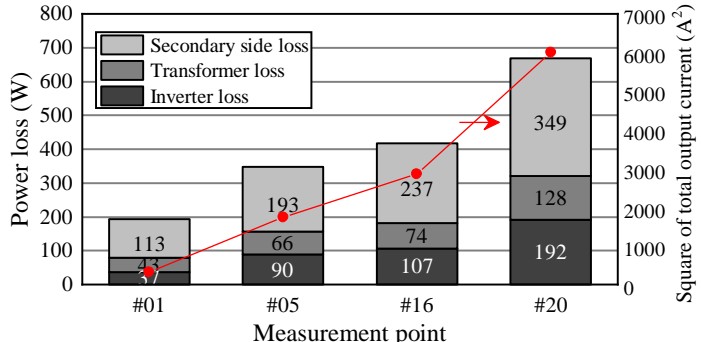

**Figure 17.** Loss analysis results.

A main reduction measure for the secondary side circuit conduction loss is the reduction in the on-resistance of the switching device. For example, S5 and S6 can be replaced with a low on-resistance device that matches the operation conditions, whereas D1 and D2 can be replaced with MOSFET that matches the operation conditions and performs synchronous rectification control. Another measure is to change the transformer turns ratio and to increase the step-down ratio, which enables devices with a lower breakdown voltage to be selected and results in a further reduction in the on-resistance. However, in this case, overly increasing the step-down ratio will lead to a reduction in the upper limit of the output power on the 48 V side.

## 5. Conclusions

In this paper, a novel dual-output isolated DC/DC converter using an integrated circuit of a current doubler and step-down chopper on the secondary side of the high-frequency transformer was proposed. Although the control method becomes more complex, the proposed circuit is expected to have a smaller size and higher efficiency compared to the existing circuits. For automotive use, a maximum output power was 2 kW and switching frequency was 400 kHz, which is a higher frequency compared to the existing studies. The objective of this study was to operate, evaluate, and identify issues for the proposed circuit with the above condition.

As a result of the actual prototype circuit test, both the voltage target response characteristics and load disturbance response characteristics were observed as designed, confirming the validity of the proposed circuit. Moreover, we identified two solutions to specific issues: (1) soft-start control at power-on and (2) a reduction in noise propagation to the output by improving the layout.

The maximum and minimum efficiencies of this circuit were 81.3 and 61.5%, respectively. Larger output power on the current doubler side and smaller output power on the step-down chopper side

resulted in higher efficiency. The loss analysis demonstrated that the loss on the secondary side accounted for more than 50%, suggesting that it is effective to reduce the on-resistance of the switching device on the secondary side to improve the efficiency.

Although we demonstrated the feasibility of the proposed circuit, we did not show that it can be compact and highly efficient as auxiliary power for EVs, PHVs, and FCVs owing to the nature of the prototype. Therefore, future tasks for improving the efficiency and downsizing include the optimization of the devices, parameters (switching device, transformer turns ratio, core material, core shape, and inductance value), and layout.

**Author Contributions:** Conceptualization, Y.M., T.N.; methodology, Y.M., T.N.; software, Y.M.; validation, Y.M., T.N.; formal analysis, Y.M.; investigation, Y.M.; data curation, Y.M.; writing—original draft preparation, Y.M.; writing—review and editing, Y.M., T.N.; supervision, T.N.; project administration, N.T., M.I. All authors have read and agreed to the published version of the manuscript.

**Funding:** This research received no external funding.

**Conflicts of Interest:** The authors declare no conflict of interest.

## Appendix A

Figure A1 indicates whole feedback loop image of the proposed circuit. $V_{fb1}$ and $V_{fb2}$, the feedback voltages of the main circuit output voltage $V_{out1}$ and $V_{out2}$, are input to FPGA. G1'~G6' are generated by FPGA and input to the gate drive circuits. Gate signals G1~G6 adjusted by isolated gate drivers and gate resistances $R_g$ in the gate drive circuits input to each gate of S1~S6 in the main circuit, respectively. S1~S4 control $V_{out1}$ ($V_{fb1}$) and S5, S6 control $V_{out2}$ ($V_{fb2}$), respectively. This is a whole feedback loop image of the proposed circuit.

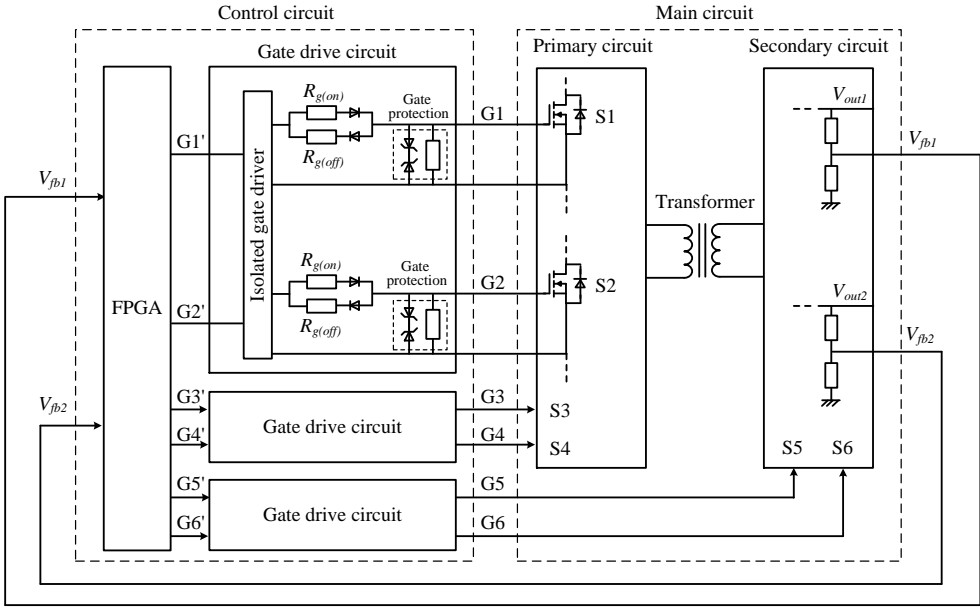

**Figure A1.** Whole feedback loop image of the proposed circuit.

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
