# Peer review of "2 kW Dual-Output Isolated DC/DC Converter Based on Current Doubler and Step-Down Chopper"

_wevj, doi:10.3390/wevj11040078_

Round 1

Reviewer 1 Report

Presented here article described a new topology for a dual-output isolated DC/DC converter. The authors presented a comprehensive solution to the problem. The topic of the paper general fits to the journal topics. The technical and scientific quality of the paper is at the good level. Despite this, the following remarks appear:

  • no complex electrical schematic (some items are shown in the fig. 5 and 7). Maybe it would be a good idea to add a complete diagram in the attachment,
  • no precise statistical information.

Reviewer 2 Report

The manuscript is dedicated to a up-to-date issue: the development of a power electronic device for use in electric vehicles.

I have no significant remarks, but I recommend revising the introduction and conclusion sections so that on the one hand the advantages and disadvantages of the proposed power scheme can be clarified in comparison with those that are widely used and on the other hand the authors' achievements stand out.
